# Aberrant splicing and defective mRNA production induced by somatic spliceosome mutations in myelodysplasia

Yusuke Shiozawa[1,2], Luca Malcovati[3,4], Anna Gallì[4], Aiko Sato-Otsubo[2], Keisuke Kataoka[2], Yusuke Sato[2,5], Yosaku Watatani[2], Hiromichi Suzuki[2], Tetsuichi Yoshizato[2], Kenichi Yoshida[2], Masashi Sanada[6], Hideki Makishima ⬥ [2], Yuichi Shiraishi[7], Kenichi Chiba[7], Eva Hellström-Lindberg[8], Satoru Miyano[7,9], Seishi Ogawa[2] & Mario Cazzola[3,4]

Spliceosome mutations are frequently found in myelodysplasia. Splicing alterations induced by these mutations, their precise targets, and the effect at the transcript level have not been fully elucidated. Here we report transcriptomic analyses of 265 bone marrow samples from myelodysplasia patients, followed by a validation using CRISPR/Cas9-mediated gene editing and an assessment of nonsense-mediated decay susceptibility. Small but widespread reduction of intron-retaining isoforms is the most frequent splicing alteration in *SF3B1*-mutated samples. *SF3B1* mutation is also associated with 3′ splice site alterations, leading to the most pronounced reduction of canonical transcripts. Target genes include tumor suppressors and genes of mitochondrial iron metabolism or heme biosynthesis. Alternative exon usage is predominant in *SRSF2*- and *U2AF1*-mutated samples. Usage of an *EZH2* cryptic exon harboring a premature termination codon is increased in both *SRSF2*- and *U2AF1*-mutated samples. Our study reveals a landscape of splicing alterations and precise targets of various spliceosome mutations.

[1] Department of Pediatrics, The University of Tokyo, Tokyo, 113-8655, Japan. [2] Department of Pathology and Tumor Biology, Kyoto University, Kyoto, 606-8501, Japan. [3] Department of Molecular Medicine, University of Pavia, 27100 Pavia, Italy. [4] Department of Hematology Oncology, Fondazione IRCCS Policlinico San Matteo & University of Pavia, 27100 Pavia, Italy. [5] Department of Urology, The University of Tokyo, Tokyo, 113-8655, Japan. [6] Department of Advanced Diagnosis, Clinical Research Center, Nagoya Medical Center, Nagoya, 460-0001, Japan. [7] Laboratory of DNA Information Analysis, Human Genome Center, The Institute of Medical Science, The University of Tokyo, Tokyo, 108-8639, Japan. [8] Department of Medicine, Center for Hematology and Regenerative Medicine, Karolinska Institutet, SE-171 77 Stockholm, Sweden. [9] Laboratory of Sequence Analysis, Human Genome Center, The Institute of Medical Science, The University of Tokyo, Tokyo, 108-8639, Japan. These authors jointly supervised this work: Seishi Ogawa, Mario Cazzola. Correspondence and requests for materials should be addressed to S.O. (email: sogawa-tky@umin.ac.jp) or to M.C. (email: mario.cazzola@unipv.it)

Myelodysplastic syndromes (MDS) and related diseases are a heterogeneous group of chronic myeloid neoplasms characterized by ineffective hematopoiesis, peripheral blood cytopenia, and an increased risk of progression to acute myeloid leukemia[1,2]. Splicing factor (SF) mutations represent a novel class of driver mutations in human cancers and affect about 50 to 60% of patients with a myeloid neoplasm with myelodysplasia[3–8]. Somatic mutations are frequently found in genes controlling 3′ splicing, including *SF3B1*, *SRSF2*, and *U2AF1*[6–8]. SF mutations are implicated in the pathogenesis of MDS most likely via abnormal RNA splicing. The mutations are heterozygous and clustered in specific amino acid residues[6–8]. These "hotspot" mutations are predicted by protein structure modeling to alter the RNA-binding specificity of the SFs[9–11]. In vitro studies using cell lines have shown that mutant *SF3B1* promotes usage of alternative branch points, leading to cryptic 3′ splice site activation[11–14]. *SRSF2* and *U2AF1* mutations were also shown in vitro to alter exon splicing patterns[9,10,15,16]. These alternative splicing patterns were confirmed in mouse models of SF mutations as well as in primary human samples[9,10,17–22]. Moreover, mouse models have shown that the mutations in *SF3B1*, *SRSF2*, and *U2AF1* cause myelodysplastic phenotypes[10,17–20].

Despite these advances, several questions remain unanswered. First, target genes of abnormal RNA splicing have not been fully elucidated. Specific alternative splicing events observed in human cells were often absent in mouse models, in which binding site sequences of mutant SFs are not always conserved[19,20,23]. This highlights the importance of identifying precise targets of aberrant RNA splicing in human hematopoietic cells. A large genetic study is required to determine the precise targets among numerous alternative splicing events observed in myelodysplasia. Second, transcriptional consequences of abnormal RNA splicing have not been systematically assessed. While the function of alternatively spliced isoforms is often difficult to predict. a frameshift or a premature termination codon (PTC) that is frequently introduced by mutant-SF-associated mis-splicing events is likely to have deleterious effects on gene function with a reduction in their canonical transcripts[10–12]. In addition, previous studies reported several aberrant isoforms that were degraded by nonsense-mediated decay (NMD) as a quality-control mechanism that eliminates transcripts harboring a PTC[10–12]. A reduction of canonical transcripts due to mutant-SF-induced mis-splicing events and their sensitivity to NMD have not been evaluated in a comprehensive manner.

To address these questions, we perform a comprehensive transcriptome analysis on a large cohort of patients with myelodysplasia. Transcriptional consequences of aberrant splicing are further assessed using NMD inhibition experiments. We identify small but widespread reduction of intron-retaining isoforms in *SF3B1*-mutated samples. Reduction of canonical transcripts is most pronounced in genes with mutant *SF3B1*-associated aberrant 3′ splice site selection. Target genes include tumor suppressors and genes of mitochondrial iron metabolism or heme biosynthesis. We also reveal that *SRSF2* and *U2AF1* mutations have a common target of splicing alterations: *EZH2*. This study provides a landscape of abnormal splicing associated with different spliceosome mutations and their precise targets.

## Results

### Mis-splicing patterns associated with spliceosome mutations.
We studied 214 patients with myeloid neoplasms with myelodysplasia followed at the Department of Hematology Oncology, Fondazione IRCCS Policlinico San Matteo, Pavia, Italy. Patient characteristics are summarized in Supplementary Table 1. The mutation status of 89 known or putative driver genes in myeloid neoplasms (Supplementary Table 2) was evaluated by targeted-capture sequencing with a mean coverage of 1009× (range 207–2306×), followed by validation using RNA sequencing or amplicon sequencing. We identified 313 non-synonymous single-nucleotide variants, 147 small insertion–deletions in the driver genes, and 170 copy number abnormalities and/or allelic imbalances (Supplementary Fig. 1). A total of 124 patients (58%) had mutations in one or more SFs, including *SF3B1* ($n = 68$ (32%)), *SRSF2* ($n = 39$ (18%)), *U2AF1* ($n = 14$ (6.5%)), and *ZRSR2* ($n = 10$ (4.7%)) (Fig. 1a).

RNA sequencing was performed on bone marrow mononuclear cells (BMMNCs; $n = 165$) and CD34+ cells ($n = 100$) obtained from 214 patients, of whom 51 were analyzed for both cell fractions (Fig. 1a). The number of unique reads ranged from 31 to 118 million, with a mean of 64 million (Supplementary Fig. 2). We identified 190,301 and 147,809 alternative splicing events in BMMNCs and CD34+ cells, respectively; these events were categorized as described in Fig. 1b. The relative expression of alternatively spliced isoforms was estimated from a "percent spliced in (PSI)" value, which reflects the fraction of reads showing alternative splicing over total reads[24,25]. To identify alternative splicing events associated with each spliceosome mutation, we compared PSI values between patients with and without the spliceosome mutations by the unequal variance *t*-test (Fig. 1c). *P*-values were adjusted for multiple testing by controlling a false-discovery rate *q*-value using the Benjamini–Hochberg method, with cut-off *q*-values <0.01.

*SF3B1* mutation was significantly associated with 3831 alternative splicing events in either BMMNCs or CD34+ cells (Fig. 1c). *SF3B1* mutation was most frequently associated with increased usage of alternative 3′ splice sites and decreased intron retention (Fig. 1d). Many of the altered 3′ splice sites were observed almost exclusively in the *SF3B1*-mutated samples; these abnormalities were commonly observed in the samples with any *SF3B1* hotspot mutation and were barely detected in the controls (Fig. 2a and Supplementary Fig. 3). Reduction of intron-retaining isoforms in *SF3B1*-mutated samples was rather unexpected. A substantial number of introns failed to be fully spliced out and were retained in SF-unmutated samples (Fig. 2a). In the *SF3B1*-mutated samples, a slight but significant decrease was observed in the relative abundance of many intron-retaining isoforms (Fig. 2a). Compared to the constitutively spliced introns, the affected introns were shorter in length (median (interquartile range (IQR)), 0.41 (0.16–1.1) vs. 1.8 (0.58–5.8) kb; $P < 0.001$) and had a higher GC content (median (IQR), 54% (44–62%) vs. 43% (38–52%); $P < 0.001$; the Mann–Whitney *U* test; Supplementary Fig. 4). No differences were found in sequences around 5′ and 3′ splice sites (Supplementary Fig. 5). We also confirmed that the proportion of sequencing reads mapped to all the intronic regions was actually decreased in *SF3B1*-mutated samples (Supplementary Fig. 6). These results indicated that *SF3B1* mutation was characterized by usage of abnormal 3′ splice sites and small but widespread reduction of intron retention.

*SRSF2* mutation was associated with 2057 events in either the BMMNC or CD34+ cell fraction (Fig. 1c). Substantially different patterns of aberrant splicing were observed in *SRSF2*-mutated and *SF3B1*-mutated patients. In contrast to *SF3B1* mutation, altered exon usage was the predominant alteration in *SRSF2*-mutated samples, accounting for 50% ($n = 1025$) of all mutant *SRSF2*-associated events (Figs. 1d and 2b). These alternative splicing events were not specific to the *SRSF2*-mutated samples, but were also observed in SF-unmutated samples, albeit less abundantly than in *SRSF2*-mutated samples (Fig. 2b). Three samples from two patients have both *SF3B1* and *SRSF2* mutations in the major clone as inferred from the variant allele frequencies (Supplementary

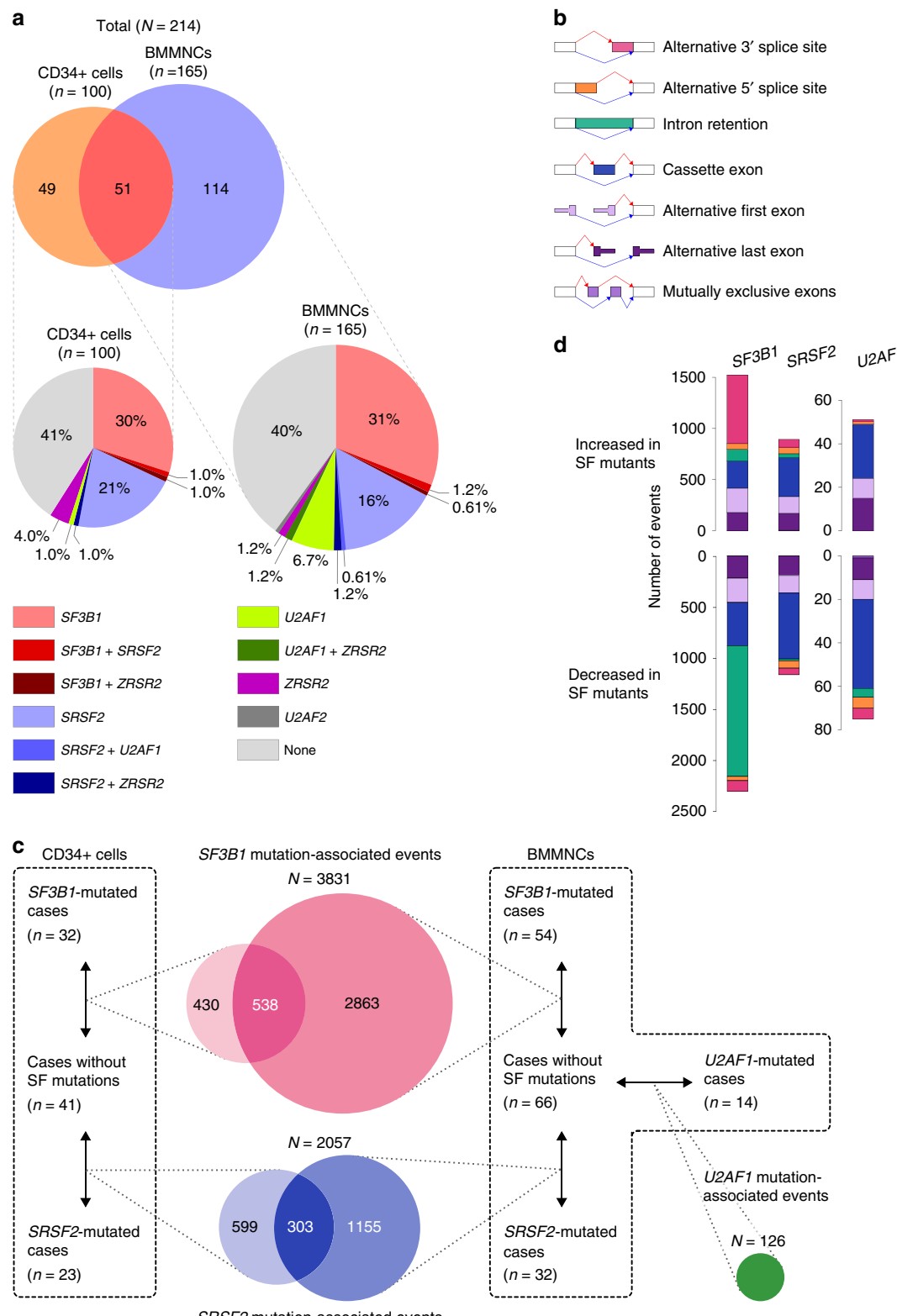

**Fig. 1** Overview of a transcriptomic analysis of 265 bone marrow samples. **a** Sources of RNA and frequencies of SF mutations. A Venn diagram shows sources of RNA. The pie charts below depict frequencies of SF mutations among 165 cases with BMMNCs and 100 cases with CD34+ cells. **b** Types of alternative splicing events. Cassette exon means inclusion or skipping of an exon. **c** An overview of differential splicing analysis. Comparison of splice junction usage between SF-mutated and non-mutated samples was performed for each cell fraction. Alternative splicing events associated with each SF mutation with a *q*-value of <0.01 were regarded as differentially spliced. Venn diagrams show numbers of alternative splicing events significantly associated with each SF mutation. **d** Types and numbers of splicing alterations significantly associated with each SF mutation. Bars in the upper half indicate alternative splicing events that were more frequently found in SF-mutated cases. Bars in the lower half are those that were more frequently found in the controls. Types of splicing alterations are color coded as in **b**

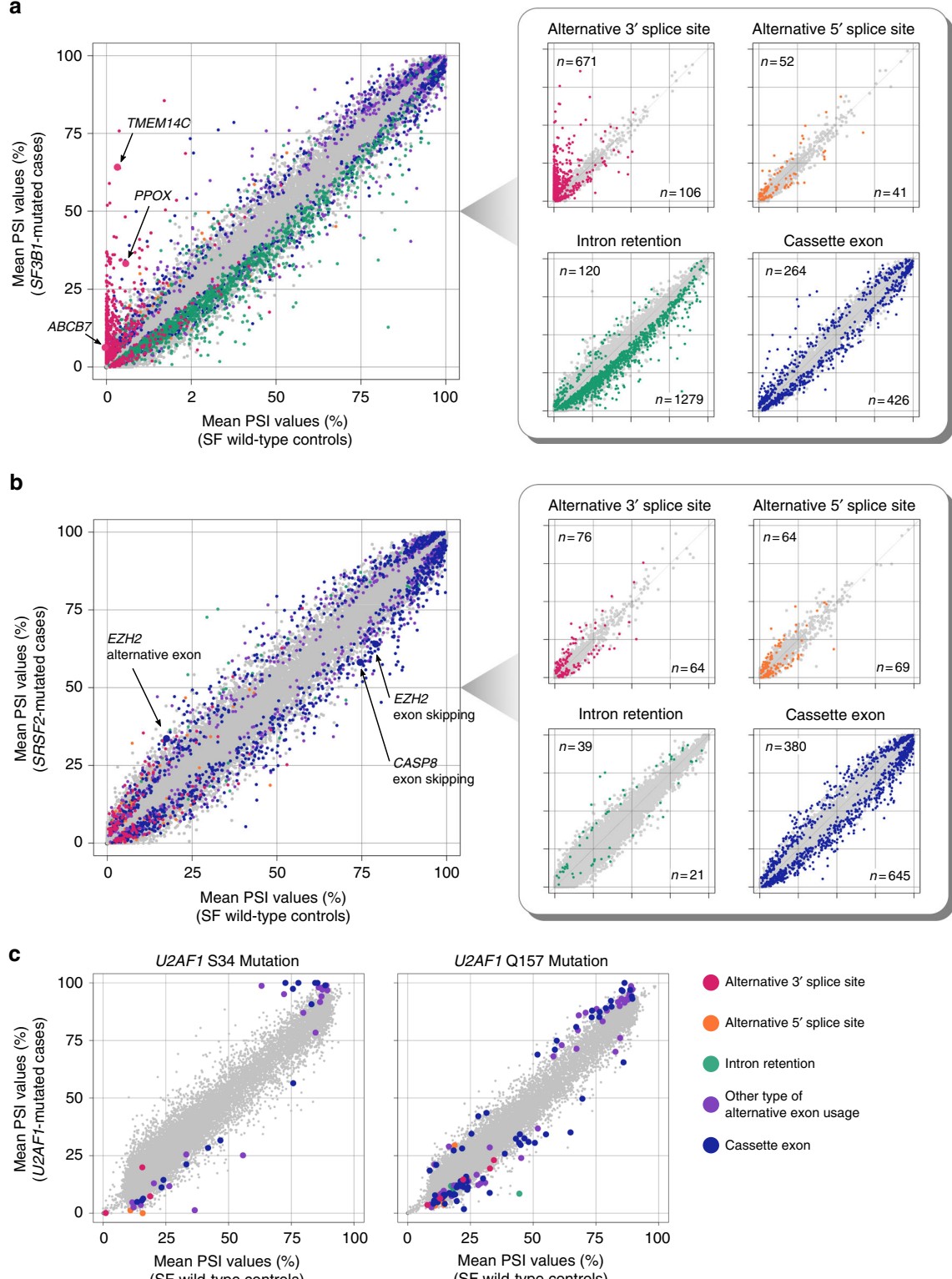

**Fig. 2** Alternative splicing events associated with spliceosome mutations. **a**, **b**, **c** Scatter plots comparing mean PSI values between the *SF3B1*- (**a**), *SRSF2*- (**b**), and *U2AF1*-mutated samples (**c**), and those without SF mutations. The number of samples in each group is shown in Fig. 1c. Significantly associated events (*q*-value <0.01) are color coded as in the bottom right panel. The panels in the right are scatter plots for each splicing pattern, also showing the number of significant events. Other types of alternative exon usage included mutually exclusive exons, alternative first exon, and alternative last exon

Fig. 7). Both *SF3B1*- and *SRSF2*-associated alterations were observed in these samples (Supplementary Fig. 8).

Similar to *SRSF2* mutation, *U2AF1* mutation was largely associated with modest changes in exon usage. Of 126 alternative

splicing events that were associated with either of the two major *U2AF1* variants involving S34 or Q157, 110 (87%) were alternative exon usage (Figs. 1d and 2c). Despite affecting the same SF, there were no substantial overlap in abnormal spicing

associated with these two hotspot mutants; only three of 126 *U2AF1*-associated alternative splicing events were affected by both mutations (Supplementary Fig. 9). This represents a difference with *SF3B1* hotspot mutations, all of which were associated with a similar set of abnormally spliced sites (Supplementary Fig. 3).

**Effect of concurrent mutations in epigenetic regulators**. Epigenetic modifications are known to have a major impact on alternative splicing[26]. We assessed the effect of mutations in epigenetic regulators that co-occurred with *SF3B1* or *SRSF2* mutations. Alternative splicing patterns were compared between each SF-mutated samples with and without co-mutations in epigenetic modifiers including *TET2, DNMT3A, IDH1/IDH2, ASXL1*, or *EZH2* (Supplementary Table 3). Although some mutations were mutually exclusive with SF mutations, a sufficient number of SF-mutated patients had concurrent mutations especially in *TET2* (Supplementary Table 3). However, we did not find any significant alterations in splicing patterns associated with co-mutations in epigenetic regulators (*q*-values <0.1 by *t*-test). We next focused on the effect of *TET2* mutations in genes that were known to be differentially methylated in *TET2*-mutated leukemia[27]. Again, no difference was found between alternative splicing patterns in SF-mutated samples with and without *TET2* mutation (Supplementary Fig. 10).

**Effects of splicing alterations at the transcript levels**. In terms of predicted proteins, alternative splicing events were classified into five groups: (i) truncating alterations, (ii) non-truncating alterations, (iii) alternative first coding exons, (iv) alternative last coding exons, and (v) alterations in non-coding regions. Truncating alterations introduce a frameshift or a PTC into a transcript, while non-truncating alterations yield an in-frame isoform with nucleotide deletion or insertion.

In accordance with previous reports, alternative 3′ splice sites associated with *SF3B1* mutation were clustered within 10–30 bp upstream of canonical 3′ splice sites, resulting in a nucleotide insertion at the authentic exon–exon junction (Supplementary Fig. 11)[11,19,28]. Of 671 alternative 3′ splice sites significantly associated with mutant *SF3B1*, 373 (56%) yielded a truncated isoform, while 192 (29%) were non-truncating events (Fig. 3a). Many *SRSF2* mutations were also associated with alternative exon usage that resulted in truncated isoforms, explaining 428 of 1025 (42%)) alternative cassette exons associated with mutant *SRSF2* (Fig. 3b).

Truncating splicing alterations are likely to have deleterious effects on gene function with a reduction in their canonical transcripts. To assess the effects of mutant-associated alternative splicing, we compared gene expression in CD34+ cells from patients with and without spliceosome mutations using the generalized linear model likelihood ratio test. The expression level of transcripts without truncating splicing alterations was estimated based on their PSI values. Target genes of mutant *SF3B1*-associated alternative 3′ splice sites showed a highly consistent reduction in canonical transcripts; 145 of 290 (50%) target genes expressed in CD34+ cells were significantly downregulated (*q*-value <0.01, Fig. 3c). Conversely, 25 of 30 genes (83%) most significantly downregulated in *SF3B1*-mutated samples (*q*-value $<1 \times 10^{-20}$) contained one or more aberrant 3′ splice sites related to mutant *SF3B1* (Fig. 3c). None of the other alterations associated with mutant *SF3B1* or *SRSF2* showed such a strong correlation with gene expression (Fig. 3d, and Supplementary Figs. 12 and 13).

**Nonsense-mediated decay of aberrant transcripts**. The marked reduction of canonical transcripts suggests that mutant-SF-associated splicing alterations represent the mechanism underlying downregulation of target genes, particularly in cells harboring mutated *SF3B*1. However, only a trace amount of aberrant transcripts was detected for some downregulated genes (Fig. 4a). We hypothesized that these aberrant transcripts were normally transcribed but were degraded by NMD (Fig. 4a). To test this hypothesis, we assessed whether the transcripts with splicing alterations increased after inhibition of NMD by paired *t*-test.

Primary bone marrow cells with mutated *SF3B1* ($n = 4$) or *SRSF2* ($n = 4$), as well as those having no known SF mutations ($n = 2$), were treated with cycloheximide (CHX) to inhibit NMD, and were then subjected to RNA sequencing, together with untreated cells. Effective inhibition was confirmed by an increase in the level of the physiological targets of NMD (Supplementary Fig. 14). We identified 6591 alternative splicing events that were specifically detected after CHX treatment, but not in the primary samples. Such events are expected to represent complete degradation by NMD under physiological conditions. However, none of them were associated with either *SF3B1* or *SRSF2* mutation (*P*-value <0.05 by *t*-test) and it is unlikely that we had failed to detect major targets of these mutations in the analysis of the primary samples. We thus focused our analysis on already identified splicing alterations. While CHX treatment did not substantially influence the PSI values of non-truncating splicing alterations, a significant (*q*-value <0.01) increase in PSI values was detected in 76 (3.9%) and 34 (5.2%) truncating events associated with *SF3B1* and *SRSF2* mutations, respectively (Fig. 4b and Supplementary Figs. 15 and 16). The most common targets of NMD were transcripts with mutant *SF3B1*-associated alternative 3′ splice sites. A significant increase after CHX treatment was detected in 48 events (Fig. 4c), which were consistently associated with a reduction in the corresponding canonical transcripts (Fig. 4d). Transcripts with other splicing patterns were also degraded by NMD (Supplementary Figs. 15 and 16), although the reduction in the corresponding authentic transcripts was less prominent compared to that in the targets of mutant *SF3B1*-associated alternative 3′ splice sites (Supplementary Figs. 12 and 13).

Overall, these results suggest that the paucity of the aberrant transcripts in some targets was a result of degradation by NMD, indicating a larger effect of the splicing alterations on gene expression than expected from the observed amounts of abnormal transcripts in the primary samples (Fig. 4a).

**Recapitulation of mutant *SF3B1*-associated abnormal splicing**. To test whether mutant *SF3B1*-associated splicing alterations were solely attributable to *SF3B1* mutation, we generated isogenic cell lines carrying the *SF3B1*[K700E] mutation. Clustered regularly interspaced short palindromic repeat (CRISPR)/CRISPR-associated protein-9 (Cas9) nuclease technology was used to knock in a heterozygous *SF3B1*[K700E] allele, where HEK293T cells were chosen as target cells because of technical feasibility. A CRISPR/Cas9-blocking synonymous mutation was also introduced to obtain *SF3B1* wild-type controls. We isolated six independent CRISPR cell clones: three with a heterozygous *SF3B1*[K700E] mutation and three with wild-type *SF3B1* (Supplementary Fig. 17).

PSI values of 3831 mutant *SF3B1*-associated alternative splicing events were compared between the CRISPR cell lines with and without the *SF3B1*[K700E] mutation (Supplementary Fig. 18). A total of 909 events (24%) showed consistent changes with a *P*-value <0.05 by *t*-test in clones with a heterozygous *SF3B1*[K700E] allele (Fig. 5a). Increased usage of aberrant 3′ splice sites was most

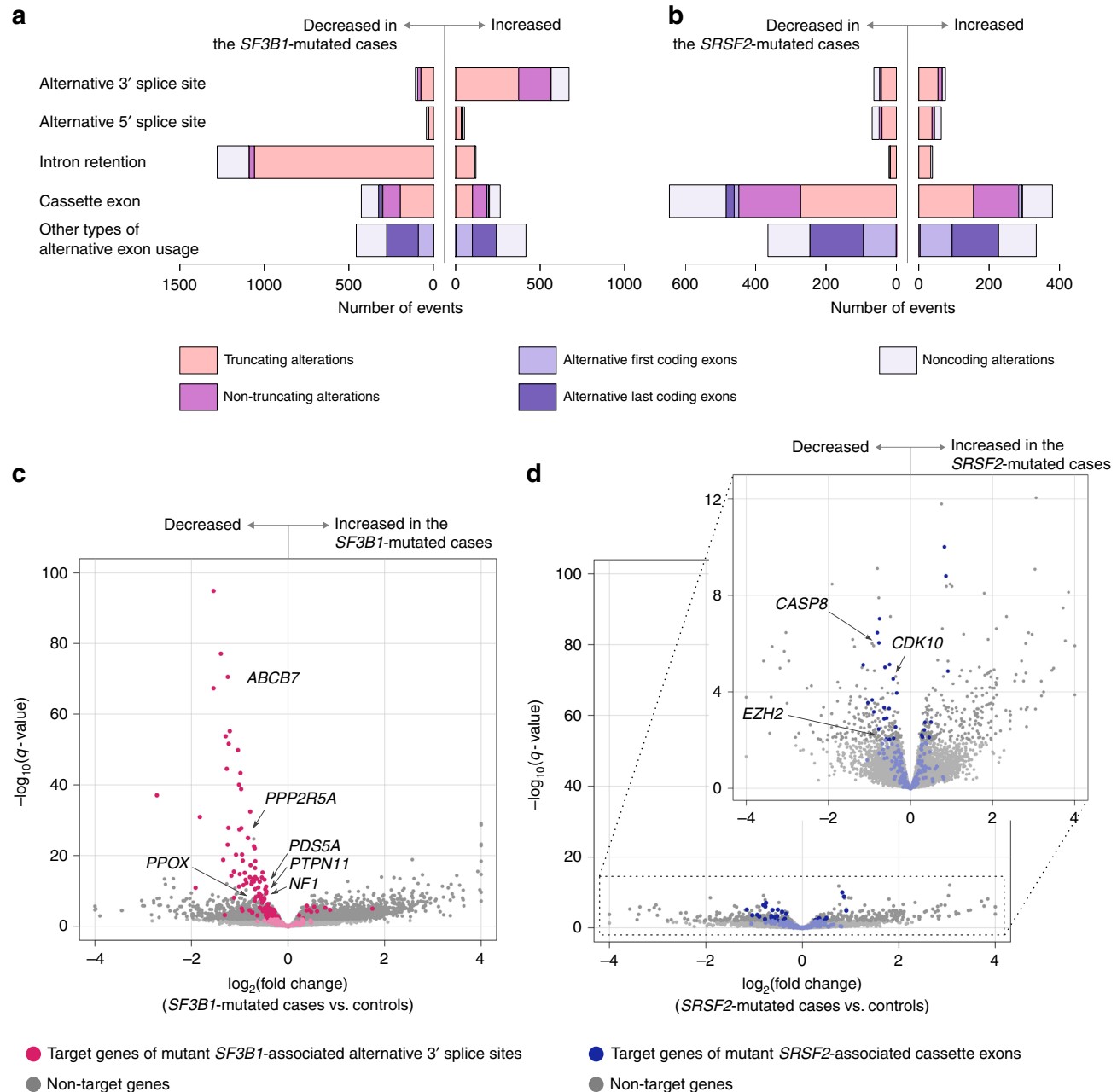

**Fig. 3** Transcriptional consequences of splicing alterations. **a**, **b** Types and numbers of splicing alterations significantly associated with *SF3B1* (**a**) and *SRSF2* mutations (**b**). Bars in the right indicate alternative splicing events that were more frequently found in SF-mutated cases. Bars in the left are those that were more frequently found in the controls. Colors indicate consequences of splicing alterations at the transcript level. **c**, **d** Volcano plots comparing expression levels of the authentic transcripts between the *SF3B1*- (**c**) and *SRSF2*-mutated CD34+ cells (**d**) and those without known SF mutations. The number of samples in each group is shown in Fig. 1c. *X*-axis indicates fold changes in gene expression on a log$_2$ scale. *Y*-axis indicates *q*-values on a negative log$_{10}$ scale. Target genes of mutant *SF3B1*-associated alternative 3′ splice sites are depicted in red (**c**). Those of mutant *SRSF2*-associated cassette exons are depicted in blue (**d**)

frequently recapitulated in *SF3B1*-mutated clones (599 of 671 events (89%), Fig. 5a, b). Many target genes of mutant *SF3B1*-induced alternative 3′ splice sites again showed a reduction in canonical transcripts (Supplementary Fig. 19). As in the primary bone marrow samples, the relative expression of intron-retaining isoforms showed modest differences in this isogenic model (Fig. 5c); the CRISPR cell clones with the *SF3B1*$^{K700E}$ mutation frequently showed decreased retention in 1279 introns of which retention was reduced in *SF3B1*-mutated primary samples (Fig. 5c, d). It was more consistent in 134 intron retention events with

highly significant reduction in *SF3B1*-mutated primary samples (*q*-value <1 × 10$^{-5}$; Supplementary Fig. 20).

To investigate the intracellular localization of intron-retaining transcripts, cytoplasmic, and nuclear RNA were separately extracted from the CRISPR cell lines. The enrichment of nuclear RNA was confirmed by the ~40S precursor rRNA peaks in the electropherogram and relative increase in small RNAs (Supplementary Fig. 21). RT-PCR was performed for several intron-retaining isoforms of which relative expression was decreased in *SF3B1*-mutated primary samples. Isoforms with retained introns

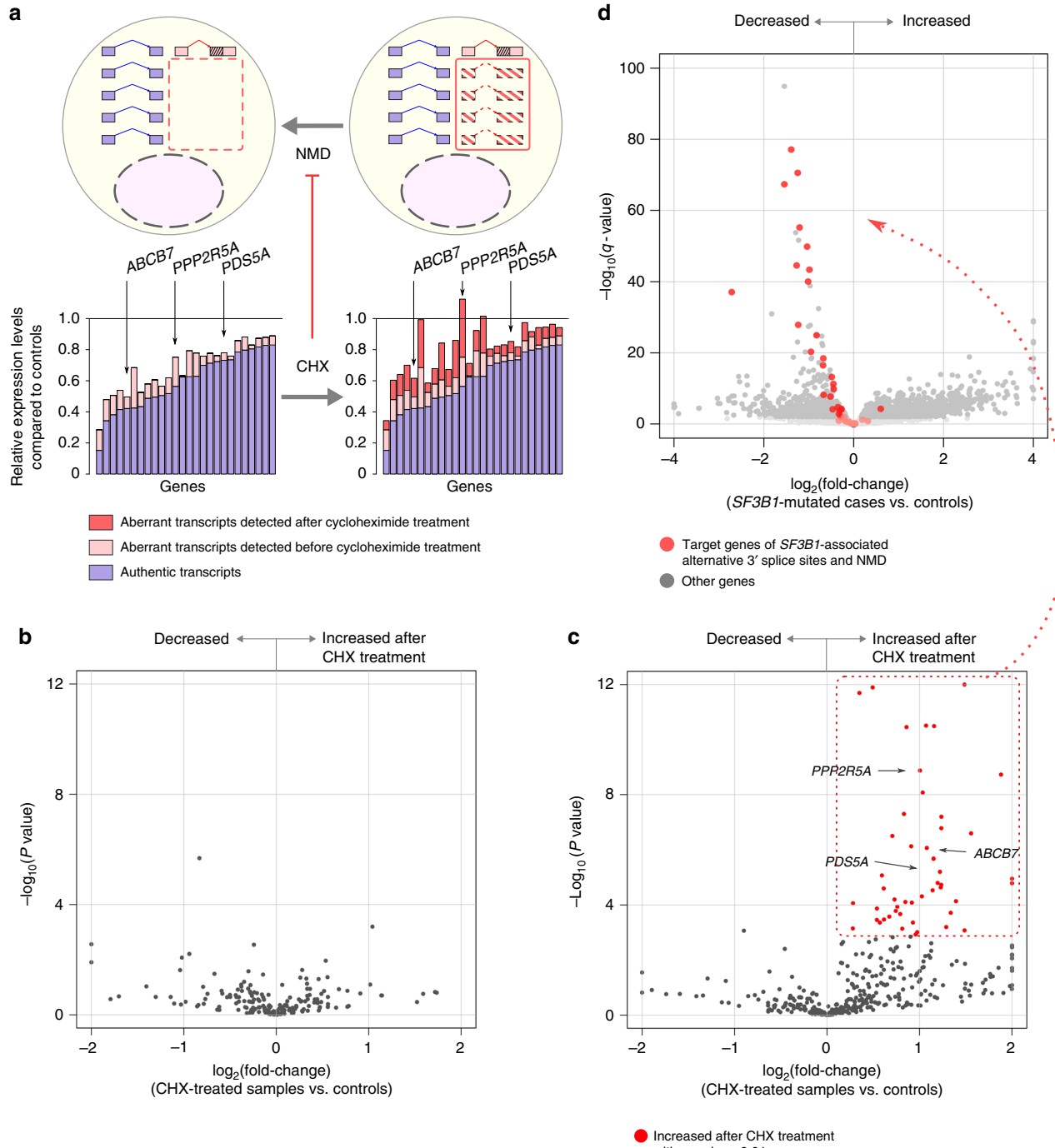

**Fig. 4** Nonsense-mediated decay of aberrant transcripts and downregulation of target genes. **a** Relative expression of some target genes of mutant *SF3B1*-associated alternative 3′ splice sites in the *SF3B1*-mutated CD34+ cells as compared with those without SF mutations. Blue bars indicate expression levels of the authentic transcripts. Pink and red bars indicate those of the aberrant transcripts detected before and after CHX treatment, respectively. These genes had a seemingly small amount of the aberrant transcripts that increased after CHX treatment. **b**, **c** Volcano plots comparing PSI values of *SF3B1*-associated splicing alterations between samples with and without CHX treatment ($n = 4$ each). Alternative splicing events were separately plotted for non-truncating events (**b**) and truncating ones (**c**). *X*-axis indicates fold changes in PSI values after CHX treatment on a $\log_2$ scale. *Y*-axis indicates *P* values on a negative $\log_{10}$ scale. Transcripts increased after CHX treatment with a *q*-value <0.01 were regarded as sensitive to NMD. **d** A volcano plot comparing gene expression levels between the *SF3B1*-mutated CD34+ cells ($n = 32$) and those without known SF mutations ($n = 41$). *X*-axis indicates fold changes in gene expression on a $\log_2$ scale. *Y*-axis indicates *q*-values on a negative $\log_{10}$ scale. Target genes of mutant *SF3B1*-associated alternative 3′ splice sites and NMD are depicted in red

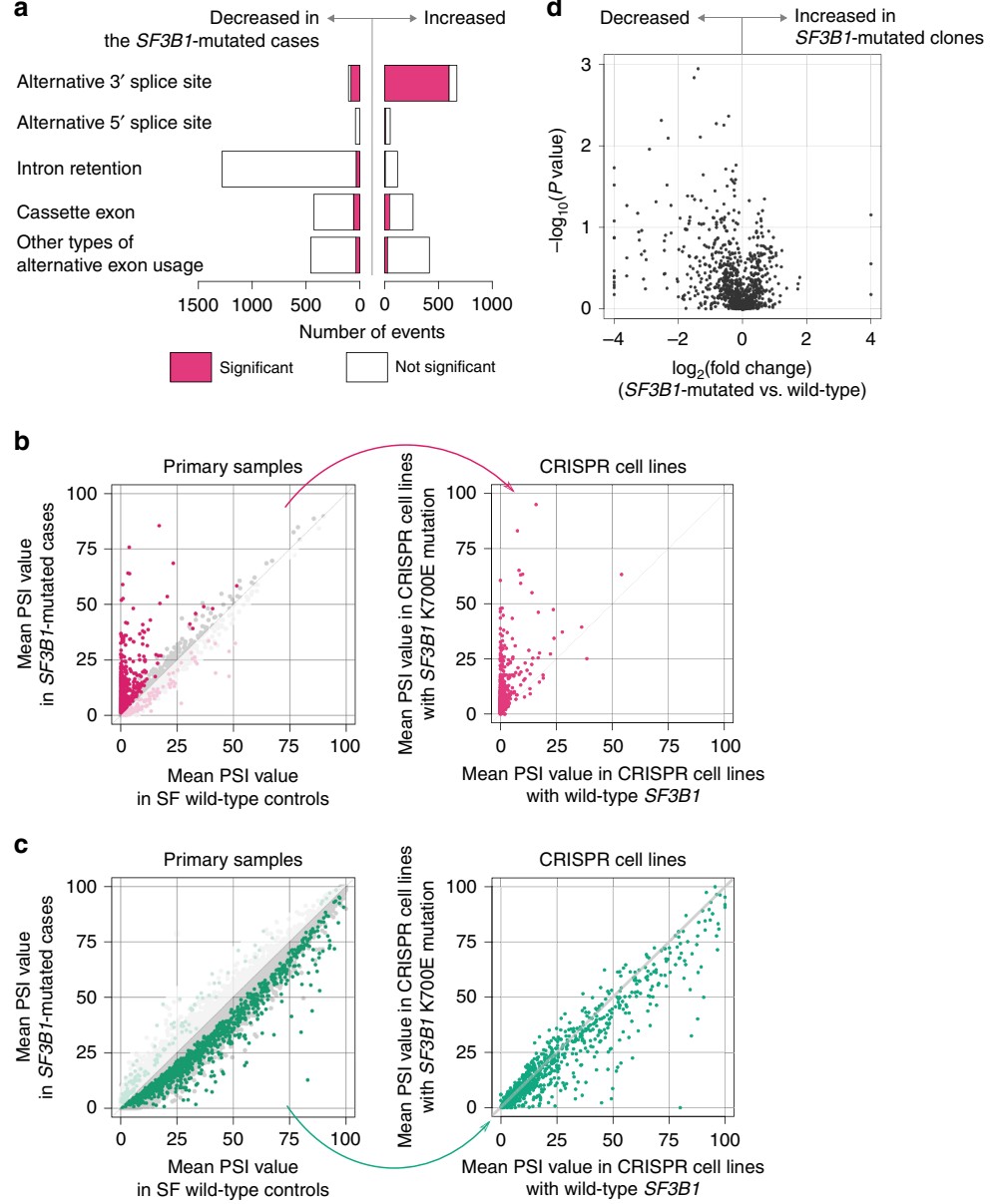

**Fig. 5** Recapitulation of mutant *SF3B1*-associated abnormal splicing in vitro. **a** The number of mutant *SF3B1*-associated events that showed a consistently significant difference in CRISPR cell lines with the *SF3B1*<sup>K700E</sup> mutation. Bars on the right indicate alternative splicing events that were more frequently found in *SF3B1*-mutated cell lines. Bars on the left represent those that were more frequently found in the controls. Red bars indicate events with consistent changes with a *P* value <0.05 in CRISPR cell lines. **b, c** Scatter plots showing mean PSI values of mutant *SF3B1*-associated alternative 3′ splice sites (**b**) and intron retention (**c**). The left and right panels show mean PSI values in primary bone marrow samples and in CRISPR cell lines, respectively. **d** A volcano plot comparing mean PSI values of mutant *SF3B1*-associated intron retentions between CRISPR cell lines with and without the *SF3B1*<sup>K700E</sup> mutation (*n* = 3 each). *X*-axis indicates fold changes in PSI values on a log₂ scale. *Y*-axis indicates *P* values on a negative log₁₀ scale

were detected not only in the nucleus, but also in the cytoplasm (Fig. 6a), indicating that some intron-retaining transcripts were normally exported to the cytoplasm. Expression levels of intron-retaining isoforms relative to spliced ones were compared between the CRISPR cell lines with and without the *SF3B1*<sup>K700E</sup> mutation. Although intron retention was decreased in both the nucleus and cytoplasm of *SF3B1*-mutated cells, the decrease was more pronounced in the cytoplasm than in the nucleus (Fig. 6a). The CRISPR cell lines were also treated with CHX to inhibit NMD, which occurs in the cytoplasm. RT-PCR of cytoplasmic RNA revealed that relative expression of intron-retaining transcripts did not increase after CHX treatment (Fig. 6b). These results indicated that a mutant *SF3B1*-associated decrease in intron retention was

not due to activated degradation of intron-retaining transcripts by NMD, but was rather caused by enhanced splicing of retained introns and/or impaired export of intron-retaining transcripts from the nucleus of *SF3B1*-mutated cells (Fig. 6c).

**The effect of transcriptional activity on abnormal splicing.** We have revealed that splicing alterations, especially those associated with *SF3B1* mutation, had a profound effect on gene expression levels. Contrary to this, transcriptional activity might influence the frequencies of alternative splicing events. To accurately assess the effect of gene expression on abnormal splicing, we compared gene expression levels with PSI values in mutant-SF-associated

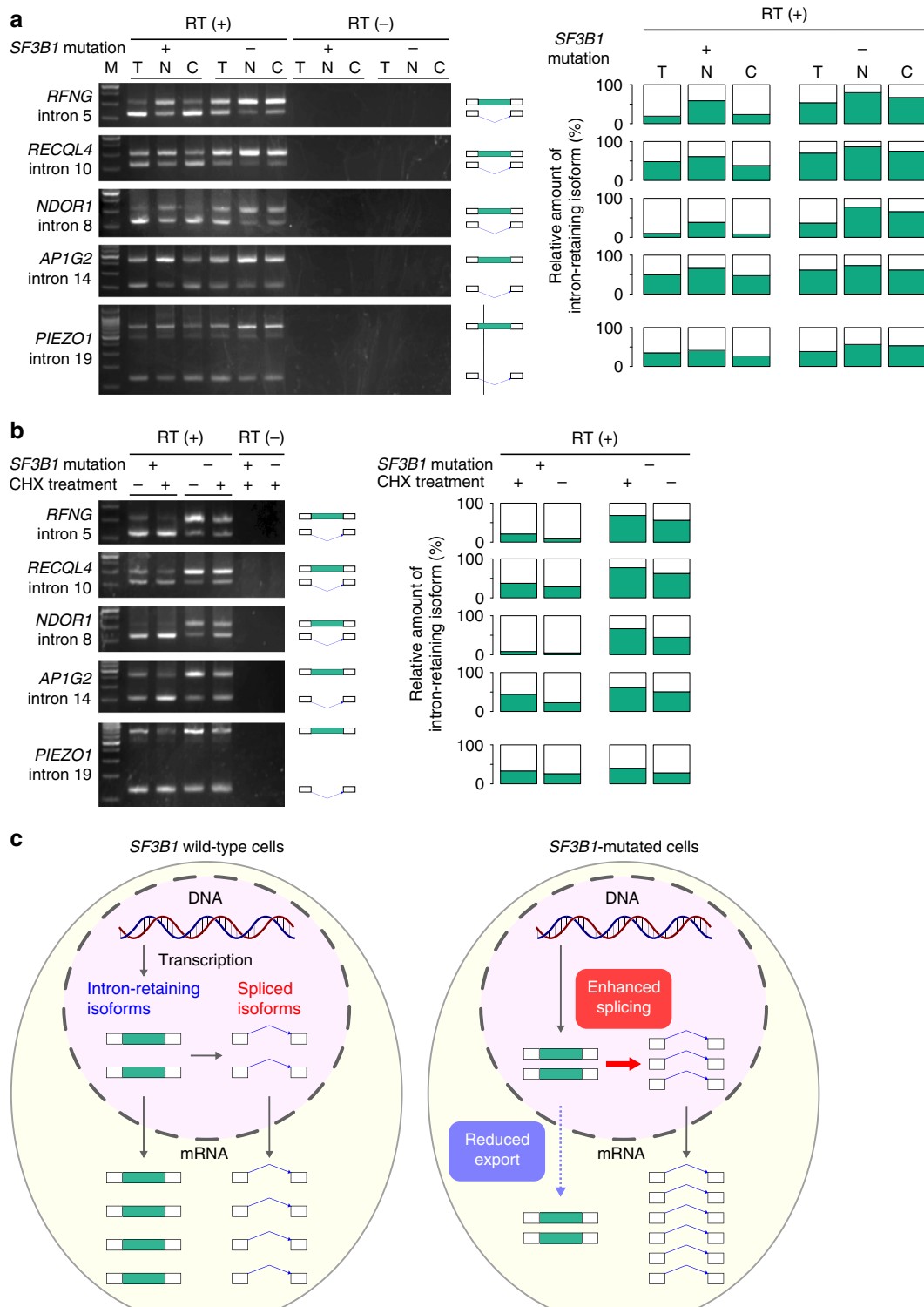

**Fig. 6** Decreased intron retention in *SF3B1*-mutated cells. **a** RT-PCR of several differentially spliced introns. RT-PCR was performed for total, nuclear, and cytoplasmic RNA extracted from CRISPR cells with or without a heterozygous *SF3B1*[K700E] allele. Samples without reverse transcription (RT) were used as a negative control. T indicates total RNA; N, nuclear RNA; C, cytoplasmic RNA. Relative amount of the intron-retaining isoforms that were calculated from the intensity of bands are also shown on the right. **b** RT-PCR of several differentially spliced introns. RT-PCR was performed for cytoplasmic RNA with or without CHX treatment. Relative amount of the intron-retaining isoforms that were calculated from the intensity of bands are also shown on the right. **c** A schematic shows the amount of intron-retaining isoforms and spliced isoforms in the nucleus and the cytoplasm. The left panel is a schematic of *SF3B1* wild-type cells, and the right is for *SF3B1*-mutated cells

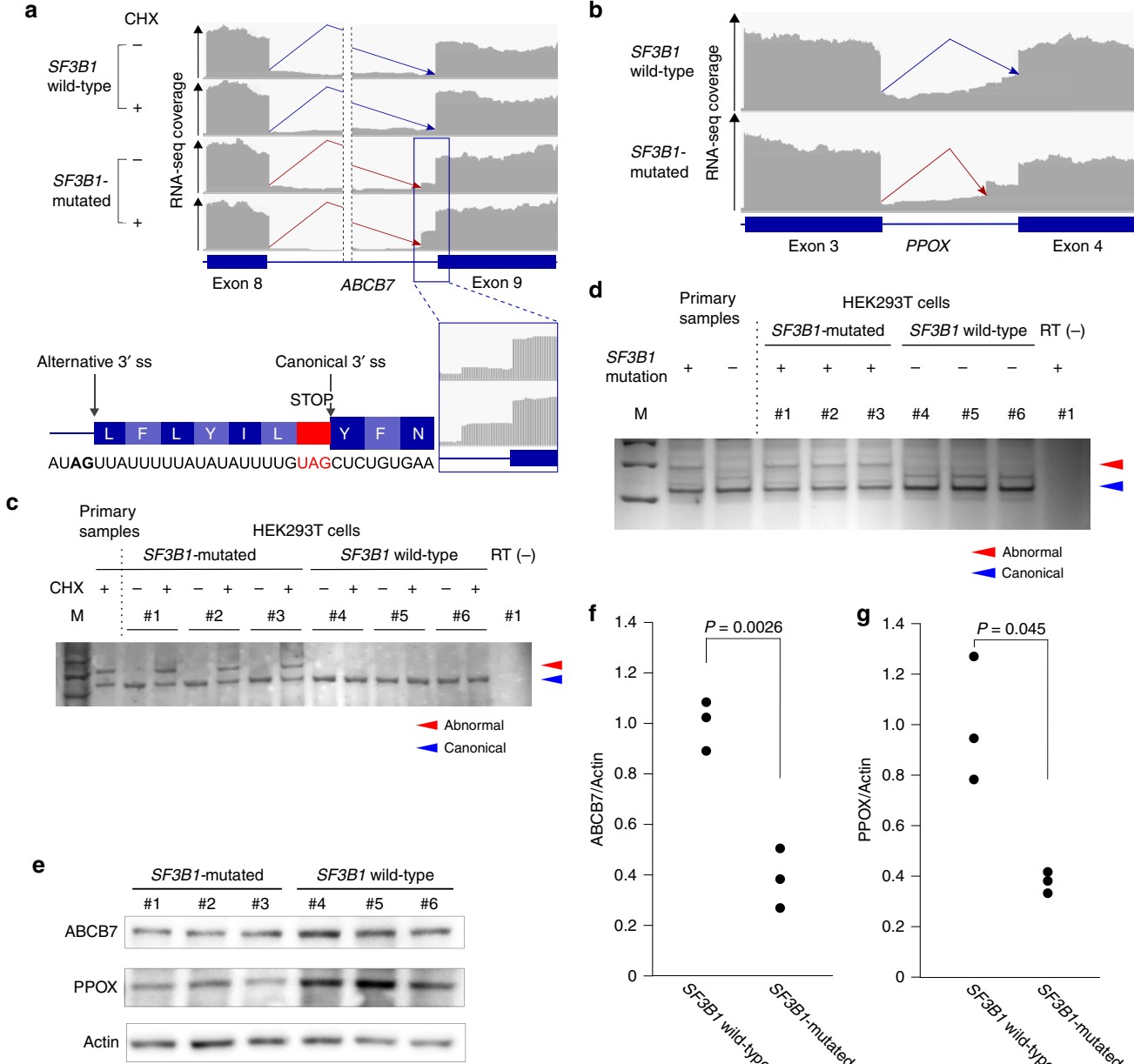

**Fig. 7** Mutant *SF3B1*-associated alternative 3′ splice sites. **a**, **b** Mutant *SF3B1*-associated alternative 3′ splice sites in *ABCB7* (**a**) and *PPOX* (**b**). RNA-seq coverage is shown for samples with mutated and wild-type *SF3B1*, as well as for those with and without CHX treatment only in **a**. The bottom figure in **a** illustrates an RNA sequence, a spliced transcript, and the corresponding amino acids near the 3′ splice site of exon nine of *ABCB7*. **c**, **d** RT-PCR of the differentially spliced sites of *ABCB7* (**c**) and *PPOX* (**d**). RT-PCR was performed for primary MDS samples and CRISPR cell lines with or without the *SF3B1*[K700E] mutation. Clones #1 to #3 carried a heterozygous *SF3B1*[K700E] allele, while clones #4 to #6 had only wild-type alleles. Samples without reverse transcription (RT) were used as a negative control. Samples treated with CHX are also shown in **c**. Canonical and abnormal transcripts are indicated by blue and red triangles, respectively. **e** Immunoblots of CRISPR cell lines with or without the *SF3B1*[K700E] mutation. Lysates were processed for immunoblotting with antibodies against denoted proteins. **f**, **g** Relative protein levels of ABCB7 (**f**) and PPOX (**g**). The dot plots show protein levels in three independent clones. Statistical significance was determined by the unequal variance *t*-test; *t*-values were 6.9 (**f**) and 4.3 (**g**)

splicing alterations that fulfilled the following criteria: (i) events almost specifically (>10-fold) detected in the SF-mutated samples; (ii) mean PSI values >10%; (iii) non-truncating events that were not sensitive to NMD. These criteria were met by 26 events, all of which were mutant *SF3B1*-associated alternative 3′ splice sites. To remove the effect of contaminating non-tumor cells, the analysis was limited to 16 samples with *SF3B1*[K700E] allele that was expressed in >90% of the cells as estimated from the variant allele frequencies in RNA: nine bone marrow CD34+ cell samples, four BMMNC samples, and three HEK293T cell lines with *SF3B1*[K700E]

mutation. As shown in Supplementary Fig. 22, no consistent relationship was found between gene expression levels and PSI values. Actually, a gene expression level was not a significant predictor of a PSI value (*P* = 0.09 by Wald test), when we modeled the effect of gene expression levels using a generalized linear model with a Gaussian distribution.

**Target genes of mutant *SF3B1*-associated abnormal splicing.** We next sought to identify pathogenic splicing alterations associated with *SF3B1* mutation. We focused on those events that

were recapitulated in the CRISPR cell lines and displayed a moderate to large effect: those with a mean PSI value that is greater than a 2-fold increase compared to controls and was greater than 10% in either primary samples or those treated with CHX. These criteria were met in 250 of 3831 (6.5%) mutant *SF3B1*-associated alterations.

*SF3B1* mutation is associated with the MDS with ring sideroblasts characterized by defects in heme biosynthesis and iron accumulation in mitochondria[29]. Of note, three genes involved in heme biosynthesis and iron metabolism showed aberrant splicing in *SF3B1*-mutated patients: *ABCB7*, *PPOX*, and *TMEM14C* (Fig. 2a). To assess whether these alterations were

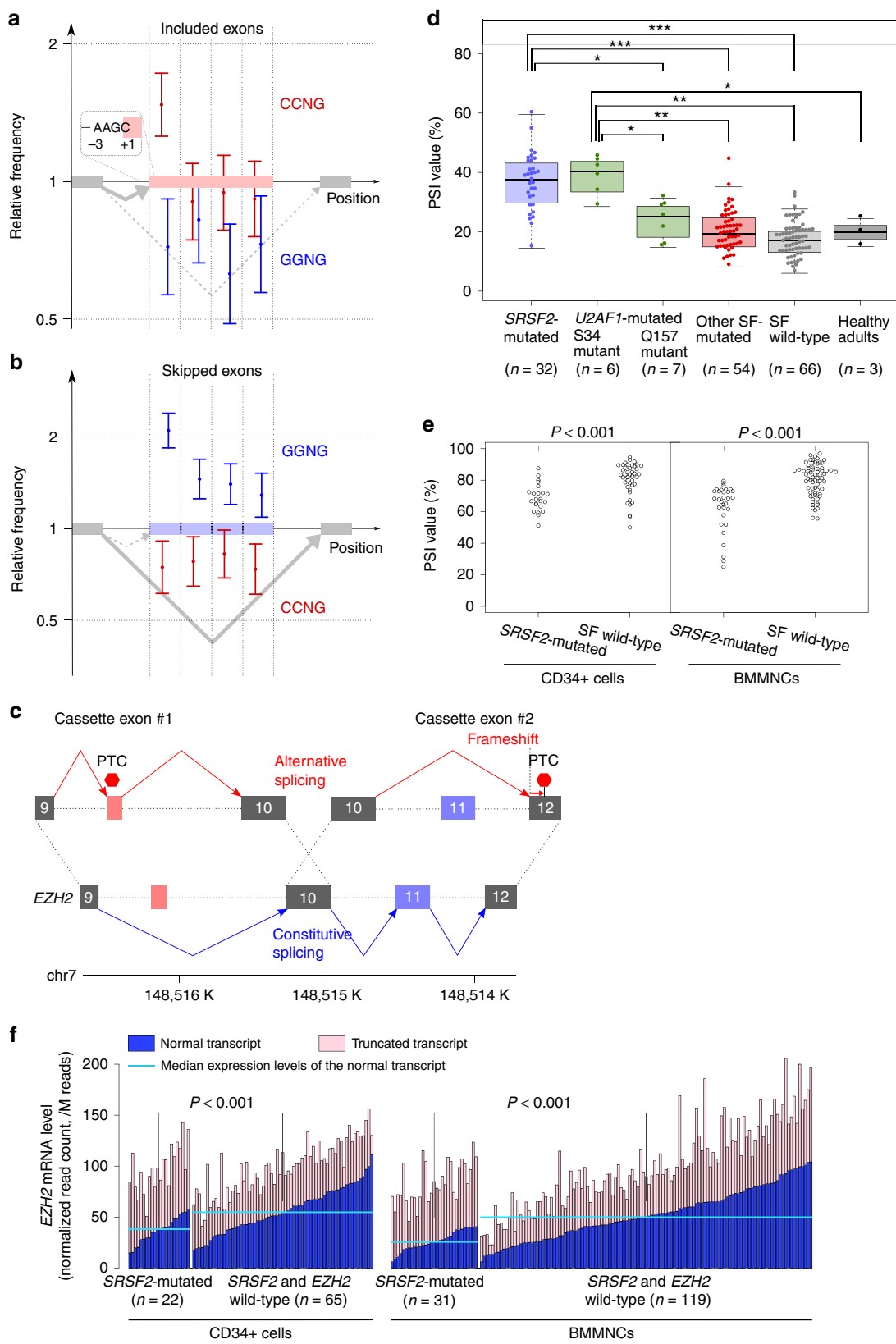

observed in erythroid cells, bone marrow CD34+ cells from eight SF3B1-mutated patients and four unmutated ones were differentiated in vitro into erythroid cells. RNA sequencing was performed on cells recovered on day 7 and day 14, which roughly corresponded to CD36+ intermediate erythroblasts and glycophorin A+ mature erythrocytes, respectively[30,31]. SF3B1-associated alternative 3′ splice sites, including those of ABCB7, PPOX, and TMEM14C, were similarly observed in the cultured erythroid cells both at day 7 and at day 14, indicating consistent effect across different hematopoietic lineages (Supplementary Fig. 23).

ABCB7, which encodes a protein involved in the transport of heme from the mitochondria to the cytosol, was markedly downregulated in the SF3B1-mutated samples as previously reported;[11,32] abnormal splicing of this gene was barely detected in the primary specimens, but became evident after CHX treatment of the SF3B1-mutated samples (Fig. 7a). The aberrant 3′ splice site was located at 21 bp upstream from the authentic junction, inserting a nucleotide sequence harboring a PTC (Fig. 7a). PPOX, which encodes an enzyme of heme biosynthesis, was another target of mutant SF3B1-associated misrecognition of 3′ splice sites that introduces a frameshift (Fig. 7b). The mutant SF3B1-associated TMEM14C transcripts also had an alternative 3′ splice site located in the 5′-untranslated region (UTR), although this neither altered the amino acid sequence nor affected gene expression as evaluated by luciferase assay (Supplementary Figs. 24 and 25). Collectively, SF3B1 mutation induces abnormal splicing and downregulation of two genes related to heme biosynthesis, ABCB7 and PPOX. The splicing abnormalities in these genes were recapitulated in HEK293T cells carrying a heterozygous SF3B1[K700E] allele introduced by CRISPR/Cas9-mediated gene editing (Fig. 7c, d). Downregulation of ABCB7 and PPOX was also confirmed at the protein level (Fig. 7e–g).

The candidate functional targets of mutant SF3B1-associated splicing also contained a number of cancer-related genes. Typical examples included known or putative tumor suppressors, such as NF1, DICER1, PML, PDS5A, MAP3K7, and PPP2R5A, in which SF3B1 mutation resulted in usage of alternative 3′ splice sites (Supplementary Fig. 26). The abnormal splicing in these genes yielded truncated transcripts with a reduction of canonical transcripts (Fig. 3c).

**SRSF2-associated mis-splicing and its functional targets.** SRSF2 is an exonic splicing enhancer that binds to a consensus SSNG sequence (where "S" denotes C or G) with no clear preference for a particular sequence within this motif[33]. SRSF2 P95 mutants were reported to have an altered affinity for this consensus sequence: an increased affinity for a CCNG motif with a decreased affinity for a GGNG motif[10]. This leads to an enhanced inclusion of exons with a CCNG sequence and a skipping of those with a GGNG sequence. We compared relative frequencies of SSNG motifs between constitutively and differentially spliced exons in SRSF2-mutated samples. In agreement with previous reports[10,16], the included exons showed enrichment of CCNG and depletion of GGNG, and vice versa for the skipped exons (Fig. 8a). We further examined the spatial distribution of SSNG motifs within the differentially spliced exons. The relevant CCNG and GGNG motifs were not uniformly distributed across the entire sequence of the affected exons, but were more enriched in their most 3′ quarter (Fig. 8b). This suggests that like SF3B1 mutants, SRSF2 mutants preferentially affect splicing at 3′ splice sites rather than at 5′ splice sites.

No cancer-related genes showed SRSF2-associated splicing alterations characterized by a PSI value greater that 10%, and a more than 2-fold increase compared to controls. Despite the modest effect of individual mutant SRSF2-associated events, EZH2 had multiple truncating splicing alterations. EZH2 encodes the catalytic component of the polycomb repressive complex 2, which acts as a tumor suppressor and is characterized by recurrent loss-of-function mutations in myeloid malignancies[34,35]. SRSF2-mutated samples showed an increased inclusion of a cryptic exon harboring a PTC between exons 9 and 10 of EZH2 (RefSeq NM_004456), as previously reported[10]. We also identified an additional EZH2 exon (exon 11) that was more frequently skipped in SRSF2-mutated samples, leading to a premature termination (Fig. 8c–e). RT-PCR of exons 9–13 and cloning of the entire coding region of EZH2 confirmed the presence of transcripts with either or both of the two alterations (Supplementary Figs. 27 and 28). These two events cooperatively contributed to downregulation of the authentic transcript of EZH2 (Figs. 3d and 8f and Supplementary Fig. 27). The functional relevance of this finding was supported by the upregulation of the targets of polycomb repressive complex 2 in SRSF2-mutated CD34+ cells, compared to those from healthy adults (Supplementary Fig. 29)[36].

We also identified additional targets of mutant SRSF2-associated splicing alterations that are potentially involved in the development of myelodysplasia, including CASP8 and CDK10. Mutant SRSF2-associated exon skipping generated a prematurely terminated transcript (Supplementary Fig. 30), leading to a reduction of the canonical transcripts of the affected genes (Fig. 3d). We found no major target genes that were commonly affected by both mutant SRSF2- and SF3B1-associated splicing alterations; none of the major target genes of SF3B1 mutation were affected by mutant SRSF2-associated alternative splicing and showed significant reduction of the canonical transcripts.

**Fig. 8** Mutant SRSF2-associated Cassette Exons in EZH2. **a, b** Forest plots show relative frequencies of CCNG and GGNG motifs within the differentially spliced exons in the SRSF2-mutated samples. Two panels show the exons that are more frequently included ($n = 380$) (**a**) and skipped ($n = 645$) (**b**) in the SRSF2-mutated samples, respectively. Comparisons were made between the differentially spliced exons and constitutive ones ($n = 72,002$). Red and blue dots indicate relative frequencies of CCNG and GGNG, respectively. Bars indicate 95% confidence intervals. Relative frequencies are shown for exons divided into four segments with equal lengths. **c** Two cassette exons in EZH2 associated with SRSF2 mutation. The left is alternative exon usage and the right is exon skipping. Gray boxes indicate constitutive exons. Red and blue boxes denote exons with increased and decreased usage in the SRSF2-mutated samples, respectively. **d** PSI values of the cryptic exon between exons 9 and 10 of EZH2. BMMNC samples with different SF mutation status are plotted separately. The middle line in each box corresponds to the median; the lower and upper boundaries of the box indicate first and third quartiles, respectively. The whiskers represent the 1.5-fold interquartile range or the maximum/minimum data point within the range. *** indicates Bonferroni adjusted P value < 0.001; **, adjusted P value < 0.01; *, adjusted P value < 0.05. **e** PSI values of exon 11 of EZH2. SRSF2-mutated samples and those without SF mutations are plotted separately for BMMNCs and CD34+ cells. The number of samples in each group is shown in Fig. 1c. **f** mRNA levels of EZH2 isoforms compared between the SRSF2-mutated samples and those without SRSF2 and EZH2 alterations. One sample with both SRSF2 and EZH2 mutations were removed from this analysis. Normal transcripts are those from an EZH2 isoform without SRSF2-associated alterations in exon usage. Truncated transcripts are those from EZH2 isoforms with at least one SRSF2-associated altered exon usage. Levels of normal transcripts were compared using the Mann–Whitney U test. W-values were 340 in CD34+ cells and 630 in BMMNCs

**U2AF1-associated mis-splicing in cancer-related genes**. Finally, we sought to identify mutant U2AF1-associated splicing alterations of pathogenic significance. No cancer-related genes were affected by 126 mis-splicing events strongly associated with U2AF1 mutation (q-value <0.01 by t-test). However, EZH2 was included among additional 237 splicing alterations associated with U2AF1 mutation with q-value <0.05. Of note, this alteration was the aforementioned cryptic exon that was associated with SRSF2 mutation (cassette exon #1 in Fig. 8c). Inclusion of this exon was increased in the samples with the U2AF1 S34 mutation, but not in those with the Q157 mutation (Fig. 8d). We next analyzed published transcriptome data of cell line experiments expressing the U2AF1 S34F mutant or wild-type U2AF1[6]. Exogenous expression of the U2AF1 S34F mutant actually enhanced inclusion of the cryptic exon of EZH2 (Supplementary Fig. 31).

We next examined the sequence flanking the 3′ splice site of the EZH2 cryptic exon. The U2AF1 S34 mutant is shown to promote recognition of C or A instead of T located three bases downstream from the 3′ splice site (−3 position), while the Q157 mutant enhances recognition of G instead of A at the +1 position[9,21,22]. This altered pattern was confirmed in the mutant U2AF1-associated cassette exons in our analysis (Supplementary Fig. 32). The 3′ splice site of the EZH2 cryptic exon has A at the −3 position, which is preferentially bound by the U2AF1 S34 mutant protein (Fig. 8a). In contrast, the +1 position of this cryptic exon was C, which does not promote binding of the U2AF1 Q157 mutant (Fig. 8a). These results were consistent with specific increase in the usage of the EZH2 cryptic exon in the U2AF1 S34 mutant.

The association between the U2AF1 S34 mutation and the splicing defect in EZH2 led us to hypothesized that the U2AF1 S34 mutation is mutually exclusive with EZH2 mutations. No mutations or deletions of EZH2 were found in six patients with the U2AF1 S34 mutation, but were found in 1 of 8 patients with the Q157 mutation in our cohort. We further included two cohorts in previous studies adding up to a total of 1325 patients with myelodysplasia[3,37]. Although 13 of 66 patients with the U2AF1 Q157 mutation had EZH2 mutations or deletions, no genetic alteration of EZH2 was found in the U2AF1 S34-mutated patients (n = 32), (P value = 0.008 by Fisher's exact test). This supports the pathogenic role of the splicing alteration of EZH2 in the U2AF1 S34-mutated myelodysplasia.

## Discussion

A high frequency of SF mutations is a distinctive feature of MDS and related myeloid neoplasms, suggesting a critical pathogenic role of deregulated RNA splicing. We performed a large-scale RNA sequencing of primary samples with different SF mutations, and investigated their impacts on RNA splicing and gene expression in a genome-wide fashion. Effects of these mutations on RNA splicing were further interrogated using NMD inhibition experiments and CRISPR/Cas9-mediated gene editing.

Spliceosome mutations were associated with thousands of alternative splicing events. As previously reported, 3′ splice site alterations were predominant in SF3B1-mutated samples, whereas SRSF2- and U2AF1-mutated samples were characterized by alternative exon usage[9–16,21,22]. However, we identified small reduction of intron-retaining isoforms as the most frequent mutant SF3B1-associated abnormal splicing events. Recent studies indicate that many introns are "actively" retained in mature mRNAs[38,39], some of which are stored in the nucleus, and then spliced and recruited to the cytoplasm when increased protein production is required, for example, during development and erythropoiesis[40,41]. In contrast to these nuclear transcripts, intron-retaining isoforms identified in our study were detected even in the cytoplasm of SF3B1 wild-type control cells.

Differentially spliced introns were shorter and had a higher GC content, which had been reported as features of introns that are difficult to splice[42]. These findings indicate that some introns are physiologically retained in mRNAs, followed by export to the cytoplasm. The vast majority of retained introns were confirmed to produce a PTC, which raises a possibility that activated NMD might be the cause of reduction in intron-retaining transcripts. However, the expression levels of these intron-retaining transcripts were not influenced by CHX treatment, making this possibility less likely. Decreased intron retention was more pronounced in the cytoplasm compared with the nucleus. This suggests an impaired export of intron-retaining transcripts from the nucleus of SF3B1-mutated cells. In addition, intron-retaining transcripts were also decreased in the nucleus of SF3B1-mutated cells, indicating enhanced splicing of retained introns. These results suggest an important role of SF3B1 in splicing regulation and the export of intron-retaining transcripts.

High statistical power due to the large sample size allowed us to identify precise targets of abnormal splicing associated with these mutations. SF3B1 mutation induced aberrant 3′ splice site usage in three genes related to heme biosynthesis and iron metabolism: ABCB7, PPOX, and TMEM14C. Mis-splicing events in ABCB7 and TMEM14C were the same as those reported previously[11–13,43]. In contrast to the splicing alteration of ABCB7 that introduced a PTC, that of TMEM14C was located in the 5′-UTR and did not have an effect on gene expression in a luciferase reporter assay. A mutant SF3B1-associated splicing alteration of PPOX was also reported previously, but it was differential exon usage, not an alternative 3′ splice site identified in our study[44]. Misrecognition of 3′ splice sites of PPOX introduced a frameshift and a PTC. Taken together, the splicing alterations in ABCB7 and PPOX were the candidates that might be responsible for ring sideroblast formation in SF3B1-mutated myelodysplasia. We also identified several cancer-related genes affected by SF3B1-associated abnormal splicing: NF1, PDS5A, DICER1, and PML.

Little overlap had been found in alternatively spliced target genes of the three most frequently mutated SFs[45,46]. We identified EZH2 as a common target of splicing alterations associated with SRSF2 and U2AF1 S34 mutations. The role of abnormal splicing of EZH2 in SRSF2-mutated MDS was controversial in prior studies[10,16]. Through extensive analysis of a large cohort, we additionally found an exon skipping event associated with SRSF2 mutation, as well as confirming enhanced inclusion of the previously reported cryptic exon[10]. We also found that one of the two alterations was associated with U2AF1 S34 mutation. This was supported by in vitro experiment data, the flanking sequence of the cassette exon that promotes binding of U2AF1 S34 mutant protein, and the mutational landscape of myelodysplasia in which EZH2 and U2AF1 S34 mutations were mutually exclusive.

Even with our large genetic study, splicing alterations were rarely found in well-known driver genes of myeloid neoplasms. It seems unlikely that the pathogenesis of SF-mutated myelodysplasia can be explained by a single mis-splicing event. Relative lack of alterations in established driver genes rather supports the previously proposed concept that multiple splicing alterations may cooperatively contribute to the pathogenesis of MDS[16]. This is paralleled by the molecular pathogenesis of the MDS with deletion 5q, in which haploinsufficiency of a combination of key genes mapping to the commonly deleted region results in a specific myelodysplastic phenotype[47]. Further studies are warranted to determine the biological effects of a single mis-splicing event, as well as a combination of multiple alterations.

## Methods

**Patients and materials**. These investigations were approved by the Ethics Committees of the Fondazione IRCCS Policlinico San Matteo, Pavia, of the Karolinska

Institutet, Stockholm, and of Kyoto University, Kyoto. Written informed consent was obtained from all patients. We enrolled 214 patients with myeloid neoplasms with myelodysplasia followed at the Department of Hematology, University of Pavia & Fondazione IRCCS Policlinico S. Matteo, Pavia, Italy. Sample size was determined to detect an absolute difference in PSI values of 10% with a two-tailed alpha of 0.001 and power of 90%. Patients' characteristics are summarized in Supplementary Table 1. Diagnostic procedures were performed according to the recommendations of the European LeukemiaNet[48]. The diagnostic criteria of the WHO classification of tumors of hematopoietic and lymphoid tissues were adopted[49,50]. Quantitative enumeration of myeloblasts, ring sideroblasts and monocytes and their precursors was performed using recently established consensus criteria[51,52]. Human bone marrow mononuclear cells (BMMNCs; StemCell Technologies, Vancouver, BC, Canada) and bone marrow CD34+ cells (Lonza, Basel, Switzerland) of three healthy adults each were used as controls.

**Targeted DNA sequencing.** Genomic DNA was available for 211 of 214 patients (99%). DNA was extracted from peripheral blood granulocytes ($n = 111$), BMMNCs ($n = 56$), bone marrow polymorphonuclear cells ($n = 43$), or bone marrow CD34+ cells ($n = 1$). Genomic DNA underwent whole-genome amplification in nine samples. Sequencing libraries were prepared from 200–1000 ng of DNA. Target capture was performed using a SureSelect custom kit (Agilent Technologies, Palo Alto, CA). RNA baits were designed to capture 89 known or putative driver genes in MDS (MDS, Supplementary Table 2) using SureDesign (Agilent Technologies). Target genes were selected based on the following criteria: (1) mutational targets reported in the large genetic studies of MDS ($n = 65$)[3–5], (2) known driver genes in myeloid neoplasms ($n = 10$)[3–5,53–63], (3) recurrently mutated genes in our cohort of patients with myeloid neoplasms with myelodysplasia ($n = 12$)[37], and (4) genes involved in the pathogenesis of anemia ($n = 2$).

Libraries were sequenced using the Illumina HiSeq 2000 or 2500 platform with a standard 100-bp paired-end read protocol (Illumina, San Diego, CA). All sequencing reads were aligned to the human reference genome (hg19) using BWA version 0.7.10. Variants of >5% allele frequency were called using SAMtools after all duplicated reads and low quality reads and bases were removed[6]. Called variants were filtered as in the previous paper and oncogenic variants were identified (Supplementary Methods)[4]. Mutational hotspots of *SF3B1* were defined as amino acid position 608–805, which corresponds to the 5th–9th HEAT repeats in the previous report[64]. Expression of these variants was examined by RNA sequencing as validation. A variant was regarded as validated if ≥2 variant allele reads were expressed. For lowly expressed genes with <10 expression reads, a mutation was confirmed by the presence of ≥1 supporting reads or amplicon sequencing. Of all, 90% and 1.2% of the variants were validated by RNA sequencing and amplicon sequencing, respectively.

Genomic copy number analysis was performed based on the sequencing depths of the target regions. The depths were calculated from the weighted sum of the fragments accounting for length and GC-biases during sequencing library amplification. The depths were compared to those of pooled controls. Genomic copy number was estimated from the obtained depth ratios using circular binary segmentation method with the DNAcopy package in R. Allelic imbalance was assessed by the allele frequencies of the heterozygous SNPs covered by >50 reads.

**RNA sequencing.** RNA was extracted from BMMNCs and/or bone marrow CD34+ cells from the patients. BMMNCs were separated from bone marrow samples by standard density gradient centrifugation, and CD34+ cells were isolated using MACS magnetic cell separation columns (Miltenyi Biotec, Bergisch Gladbach, Germany) according to the manufacturer's recommendations[65]. Total RNA was extracted with TRIzol reagent (Life Technologies, Carlsbad, CA) and treated with DNase I (Qiagen, Hilden, Germany) using RNeasy Mini Kit (Qiagen) following the recommendations of the manufacture. RNA integrity was examined with the TapeStation (Agilent Technologies), and RNA samples with RNA integrity number of >7 were sequenced. RNA sequencing was performed on BMMNCs ($n = 165$) and CD34+ cells ($n = 100$) obtained from 214 patients, of whom 51 were analyzed for both cell fractions (Fig. 1a). The RNA sequencing libraries were prepared from polyA-selected RNA using the NEBNext Ultra RNA Library Prep kit for Illumina (New England BioLabs, Ipswich, MA). Libraries were sequenced using the Illumina HiSeq 2000 or 2500 platform with a standard 100-bp paired-end read protocol.

**RNA sequencing analysis.** The sequencing reads were aligned to the human reference genome (hg19) using the RNA-seq unified mapper (RUM) version 2.0.4 and the transcripts were assembled using Cufflinks version 2.1.1[66,67]. The assemblies were merged using Cuffmerge for all the CD34+ cell samples and for all BMMNC samples[67]. All genomic coordinates are based on GRCh37/hg19.

Alternative splicing analysis was performed using JuncBASE version 0.6[24]. JuncBASE was run with default settings to identify alternative splicing events and calculate PSI values, which reflected fraction of reads showing alternative splicing. Intron retention was assessed only for introns from the RefSeq database. Alternative 5′ and 3′ sites were limited to those located within 500 nucleotides from constitutive ones. Detection of coordinate cassette exon was omitted due to computational burden. Differential splicing analysis was performed using "compareSampleSets.py" of JuncBASE with the following parameters: –thresh

5 –mt_correction BH –which_test t-test –delta_thresh 0. All tests were two-sided and equal variance was not assumed. Correction for multiple testing was done by the Benjamini–Hochberg method, with cut-off *q*-values <0.01. Differences in PSI values between samples with and without CHX treatment were tested by Cochran–Mantel–Haenszel test. Inclusion of the two cassette exons of *EZH2* in a single RNA molecule was assessed from sequence fragments spanning the two exon–exon junctions. Sequence logos were generated using WebLogo[68].

Differential expression analysis was conducted using edgeR version 3.6.8[25]. We only analyzed genes expressed at >1 counts per million (CPM) in at least six samples. Read counts derived from transcripts without truncating splicing alterations were estimated from PSI values of the events with the highest fold changes compared to the controls. Read counts were normalized using weighted trimmed mean of M-values. For downstream analyses, normalized read counts were converted to CPM and $\log_2$ transformed. Generalized linear models were used to compare gene expression data among groups. Correction for multiple testing was done by the Benjamini–Hochberg method. Genes with a false-discovery rate *q*-value <0.01 were considered significantly expressed. The R package GOseq version 1.18.0 was used to find pathways enriched among differentially expressed genes[69]. A gene set of the polycomb repressive complex 2 targets was adopted from the previous report (Supplementary Methods)[36]. Enrichment scores were calculated based on signal-to-noise ratio to depict running sum plots of the scores[70].

**CRISPR guide RNA vector construction and clone isolation.** The human *SF3B1* CRISPR guide RNA (gRNA) (5′-TGGATGAGCAGCAGAAAGTTcgg-3′) was designed using the CRISPR design tool (http://crispr.mit.edu) and cloned into the pSpCas9(BB)-2A-GFP (PX458; Addgene) vector. The 100-nt single-stranded oligodeoxynucleotide (ssODN) repair templates (Invitrogen) were designed with the flanking sequences centered on the predicted CRISPR/Cas9 cleavage site that contained CRISPR/Cas-blocking mutations with or without *SF3B1* K700E mutation. CRISPR/Cas-blocking synonymous mutation was selected based on codon-usage of human *SF3B1* by changing the arginine codon to another codon already used in the same mRNA. *SF3B1* gRNA vector along with either wild type (5′-TA GTTAAAACCTGTGTTTGGTTTTGTAGGTCTTGTGGATGAGCAGCAGA AAGTTCGTACCATCAGTGCTTTGGCCATTGCTGCCTTGGCTGAAGCAGC AA-3′) or K700E mutant (5′-TAGTTAAAACCTGTGTTTGGTTTTGTAG GTCTTGTGGATGAGCAGCAGGAAGTTCGTACCATCAGTGCTTTGGCC ATTGCTGCCTTGGCTGAAGCAGCAA-3′) ssODN was electroporated into HEK293T cells using the NEON transfection system (Invitrogen). HEK293T cells were obtained from the RIKEN Cell Bank. Cell lines were authenticated by short tandem repeat DNA profiling and routinely tested for mycoplasma infection. After 24 h, cells were sorted for expression of GFP using the BD FACSAria III cell sorter (BD Biosciences). The sorted cells were plated on 96-well plates for single-clone isolation. Positive single clones were selected and confirmed by PCR amplification (forward primer: 5′-GAGGTACACACACAGCCTGTC-3′; reverse primer: 5′-CA AGAAAGCAGCCAAACCCTAT-3′) and sequencing.

**Off-target analysis.** Gene edited HEK293T cell lines were tested for off-target editing events predicted by the Zhang laboratory CRISPR design tool (http://crispr.mit.edu) and the COSMID (http://crispr.bme.gatech.edu). The top five non-overlapping predicted off-target sites from each tool were examined (Supplementary Table 4). The region surrounding each off-target site was PCR-amplified, sequenced and confirmed not to have off-target mutations.

**RT-PCR.** Total RNA was isolated with the RNeasy Mini Kit (Qiagen). Cytoplasmic and nuclear RNA were extracted with the cytoplasmic and nuclear RNA purification kit (Norgen Biotek Corp., Thorold, Canada). RNA was reverse-transcribed using ReverTra Ace (Toyobo, Osaka, Japan) according to the manufacturer's instructions. Alternative spliced regions of *ABCB7*, *PPOX*, *EZH2*, *RFNG*, *RECQL4*, *NDOR1*, *AP1G2*, and *PIEZO1* cDNA were amplified. PCR products were run on a 4% agarose gel or a 10% or 15% polyacrylamide gel (BioRad, Hercules, CA). Uncropped gel images are shown in Supplementary Figs. 33 and 34. Images were minimally level adjusted using Adobe Photoshop Elements version 13. The relative intensity of each band was measured using ImageJ software (National Institute of Health, Bethesda, MD). Primer sequences used in RT-PCR are listed in Supplementary Table 5.

The entire coding region of *EZH2* was also PCR amplified from cDNA obtained from a patient with *SRSF2* mutation and *SRSF2* wild-type one. PCR was performed using the primes listed in Supplementary Table 5. Isolated clones were then cloned into pUC19, followed by capillary sequencing.

qRT-PCR reactions were performed with Roche LightCycler 480 SYBR Green I Master as recommended by the manufacturer, and run on a LightCycler 480 (Roche). Ct values were normalized to the housekeeping gene β-actin. Primer sequences used in qRT-PCR are listed in Supplementary Table 5.

**Immunoblots.** HEK293T cells were lysed in RIPA buffer (Santa Cruz Biotechnology, Inc., Santa Cruz, CA), subjected to SDS-PAGE and transferred to a polyvinylidene difluoride membrane (Millipore, Darmstadt, Germany). Each blot was incubated with the antibodies described in Supplementary Table 6, and signals were detected with Immobilon Western Chemiluminescent HRP Substrate

(Millipore). Uncropped images are shown in Supplementary Fig. 35. Images were minimally level adjusted using Adobe Photoshop Elements version 13.

**Erythroid cell culture**. Erythroid cells were obtained in vitro from selected CD34+ cells. Following separation, CD34+ cells were cultured ($0.5–1 \times 10^5$/mL) for 14 days in Iscove's medium supplemented with 15% BIT9500 serum substitute (Stem Cell Technologies, Tebu-Bio, Milan, Italy), 100 U/ml penicillin-streptomycin and 2 mM L-glutamine, rh-IL-3 (10 ng/mL) (PeproTech House, Tebu-Bio, Milan Italy), rh-IL-6 (10 ng/mL) (PeproTech House), rh-SCF (25 ng/ml) (PeproTech House). Epo (2 iu/ml) was added to the medium at day 7, and fresh medium supplemented as above (plus Epo) was added at day 9 and 11. Cells recovered on day 7 and day 14 were suspended in TRIzol Reagent (Life Technologies) and subjected to RNA sequencing.

**Inhibition of nonsense-mediated mRNA decay by CHX**. CHX, a translation elongation inhibitor, was used to block nonsense-mediated mRNA decay and to identify abnormal transcripts. Optimal concentration and exposure were determined after treatment with CHX (Sigma-Aldrich, Saint Louis, MO, USA) at increasing doses of 10, 20, 40, and 70 μg/mL for 1, 2, or 12 h. BMMNCs and CD34+ cells were seeded in complete medium (IMDM supplemented with 15% BIT9500, 100 U/ml penicillin-streptomycin, 2 mM L-glutamine, 10 ng/mL rh-IL-3, 10 ng/mL rh-IL-6, 25ng/ml rh-SCF) at $0.1 \times 10^6$ cells/mL density in humidified atmosphere of 5% $CO_2$ at 37 °C for 2 h. After incubation, cells were cultured at the final concentration of 70 μg/mL CHX for 2 h. Cell viability was assessed by flow cytometry using 7-amino-actinomycin D. HEK293T cells were treated with CHX at the final concentration of 100 μg/mL for 3 h. Total RNA was isolated using TRIzol Reagent (Life Technologies) and subjected to mRNA isoforms expression analysis.

**5′-rapid amplification of cDNA end**. 5′-rapid amplification of cDNA end (RACE) was performed using SMARTer RACE 5′/3′ kit (Takara) to obtain the sequence of *TMEM14C* 5′-UTR. Primer sequences used in 5′-RACE are listed in Supplementary Table 5. The products with and without the mutant *SF3B1*-associated splicing alteration were cloned into pGL3-Basic vector (Promega) directly upstream of the coding region of firefly luciferase for use in subsequent luciferase assays.

**Luciferase assay**. pGL3-Basic vector with *TMEM14C* 5′-UTR with or without the mutant *SF3B1*-associated splicing alteration, and empty pGL3-Basic construct were transfected into HEK293T cells. HEK293T cells were collected 24 h after transfection and assayed for luciferase activity using the Dual-Luciferase Reporter Assay System (Promega) and a Multilabel Plate Reader ARVO X5 (PerkinElmer, Waltham, MA). Firefly luciferase activity was normalized against Renilla luciferase activity (phRL-TK vector, Promega) in each sample and is presented relative to the activity in mock-transfected cells.

**Statistical analysis**. Numerical variables were summarized by median and range; categorical variable were described with count and relative frequency (%) of subjects in each category. Comparison of numerical variables between groups was carried out using a nonparametric approach (Mann–Whitney U test or Kruskall Wallis ANOVA). Comparison of the distribution of categorical variables in different groups was performed with the Fisher exact test.

## Data availability

Genome data that support the findings of this study have been deposited at the European Genome-phenome Archive, which is hosted at the EBI, under accession number EGAS00001002346. All other remaining data are available within the Article and Supplementary Files, or available from the authors upon request.

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

## Acknowledgements

This work was supported by the Project for Development of Innovative Research on Cancer Therapeutics (P-DIRECT, 16cm0106501h0001), Practical Research for Innovative Cancer Control (15Ack0106014h0002, 16ck0106073h0003), and Project for Cancer Research and Therapeutic Evolution (P-CREATE, 16cm0106501h0001) from Japan Agency for Medical Research and Development (AMED) to S.O., Japan Society for the Promotion of Science (JSPS) KAKENHI (15J02911 to Y. Shiozawa; 15H05668 to K.Y.; 26115009 to M.S.; 22134006, 26221308, and 15H05909 to S.O.), research grants from Associazione Italiana per la Ricerca sul Cancro (AIRC, Special Program Molecular Clinical Oncology 5 per Mille, project 1005 to M.C.; IG 15356 and 20125 to L.M.) and from Fondazione Regionale Ricerca Biomedica (FRRB, project no. 2015-0042 to M.C.), those from the Swedish Cancer Society and the Research Council of Sweden to E.H.-L., and those from the Uehara Memorial Foundation to K.Y. This research used computational resources of the Human Genome Center, the Institute of Medical Science, The University of Tokyo, Japan, and those of the K computer provided by the RIKEN Advanced Institute for Computational Science through the HPCI System Research project (hp160219). We are grateful to all the patients for their participation in this study.

## Author contributions

Y. Shiozawa performed sequencing experiments, developed bioinformatics pipelines, performed sequencing data analyses, performed functional assays, generated the figures and tables, and wrote the manuscript; L.M. collected the specimens, performed functional assays, generated the figures and tables, and wrote the manuscript; A.G. collected the specimens and performed functional assays; A.S.-O. performed sequencing experiments, sequencing data analyses, and functional assays; K.K. performed functional assays; Y. Sato performed sequencing experiments; Y.W. performed functional assays; H.S. developed bioinformatics pipelines; T.Y. developed bioinformatics pipelines; K.Y. performed sequencing experiments; M.S. performed functional assays; H.M. performed functional assays; Y. Shiraishi developed bioinformatics pipelines; K.C. developed bioinformatics pipelines; E.H.-L. collected the specimens; S.M. developed bioinformatics pipelines; S.O. co-led the entire project, generated the figures and tables, and wrote the manuscript; and M.C. collected the specimens, co-led the entire project, generated the figures and tables, and wrote the manuscript.

## Additional information

**Competing interests:** The authors declare no competing interests.

