## [Peer Review File · Nature Communications]

Reviewers' comments:

Reviewer #1 (Remarks to the Author):

The authors have adequately addressed my comments by performing additional analyses of their data and making relevant changes to the manuscript, in order to enhance its novelty and authority with some success. In addition, some of their negative findings (e.g. the fact that co-mutations and expression levels do not significantly alter the observed mis-splicing) are valuable for those investigating the mechanism of action of splicing gene mutations in MDS.

Reviewer #2 (Remarks to the Author):

Shiozawa and colleagues have performed RNA-seq on 265 bone marrow samples from 214 patients with myeloid malignancies (of which, 124 harbor splicing factor mutations) and characterized the pattern of alternative splicing. Many of the central findings have been previously reported, but several observations are novel and interesting. These include the unexpected modest but widespread decrease in intron-containing transcripts in SF3B1 mutated samples, the decreased expression of alternatively-spliced transcripts in SF3B1 mutated cases, and inclusion of the EZH2 'poison exon' previously reported in SRSF2 mutated cases also observed here for U2AF1 codon 34 mutated cases.
Comments:

1. The failure to identify any overlap in transcripts that are aberrantly spliced in SF3B1, SRSF2, and U2AF1 cases leaves a central question without a mechanistic explanation. Why are these mutations largely mutually exclusive? Furthermore, although the authors have identified consistent splicing changes associated with each genotype, what evidence have they provided to causally connect splicing changes to MDS phenotypes? The statement in the Discussion that 'multiple splicing alterations appear to cooperatively contribute to the pathogenesis of MDS' is speculative.

2. The legend for Figure 5a requires some clarification. My interpretation is that the bars represent SF3B1-associated aberrant splicing detected in patient samples and the shading reflects the fraction with support in the engineered cell line. 'Validated' is somewhat misleading, since this generally implies confirmation using orthogonal technologies. In panels b and c, the splicing changes are qualitatively recapitulated in the cell lines, but it would be useful to know how consistent the results were at the individual transcript level. Perhaps a supplemental figure could show delta PSI values for aberrant transcripts in patient samples vs. cell lines.

3. The RT-PCR image quality is not adequate in Figure 6. The results would also be more convincing with quantitative analysis of transcript ratios.

Minor:

1. The PSI scale in Supp Figure 3 is not shaded.
2. Are the x-axis labels reversed in Figure 7 panels f and g?
3. Legend for Figure 8 has panel f mislabeled.

REVIEWER 1

The authors have adequately addressed my comments by performing additional analyses of their data and making relevant changes to the manuscript, in order to enhance its novelty and authority with some success. In addition, some of their negative findings (e.g. the fact that co-mutations and expression levels do not significantly alter the observed mis-splicing) are valuable for those investigating the mechanism of action of splicing gene mutations in MDS.

We deeply appreciate the reviewer's help in improving our manuscript.

REVIEWER 2

reviewer #2 also noted in the confidential comments to the editor that it would be worthwhile for the authors to highlight the novel findings within the manuscript - so if you could please highlight what findings are new and what findings are consistent with prior reports that would be appreciated.

We are grateful to the reviewer for giving us an opportunity to improve our paper. We revised the Abstract and the Discussion section to highlight the novel findings.

Results that were consistent with prior reports are indicated as such.

Major comments

Point 1: The failure to identify any overlap in transcripts that are aberrantly spliced in *SF3B1*, *SRSF2*, and *U2AF1* cases leaves a central question without a mechanistic explanation. Why are these mutations largely mutually exclusive? Furthermore, although the authors have identified consistent splicing changes associated with each genotype, what evidence have they provided to causally connect splicing changes to MDS phenotypes? The statement in the Discussion that ‘multiple splicing alterations appear to cooperatively contribute to the pathogenesis of MDS’ is speculative.

We appreciate the reviewer’s suggestion. We have not performed experiments that assessed the effect of each splicing alteration on hematopoiesis, which will require well-designed experiments. In addition, experiments are required for many putative driver events. It would be grateful if the reviewer allows us to address these questions in the future research. Molecular basis of the mutually exclusive nature of splicing factor mutations cannot also be revealed by our genetic study, but requires experimental investigation. Although two patients in our cohort had both *SF3B1* and *SRSF2* mutations, the number of patients was too small to assess the effect of multiple splicing factor mutations. This is also an essential question that should be studied in future experiments.

Cooperative contribution of multiple splicing alterations to the pathogenesis of MDS is proposed in a previous paper (Zhang J *et al.* PNAS. 2015;112:E4726-34).

Even with the high statistical power of our large genetic study, few well-known driver genes of myeloid neoplasms were identified as a target of splicing alterations. We speculated that this result supports the above concept and would be better to be discussed in the text. We have corrected the sentences to make it clear that the statement is only speculative and have also cited the previous paper. However, if such speculation is better to be avoided, we welcome the editors to decide whether this paragraph is included in the manuscript.

5th paragraph of the Discussion section on Page 17, in Line 24

Even with our large genetic study, splicing alterations were rarely found in well-known driver genes of myeloid neoplasms. It seems unlikely that the pathogenesis of SF-mutated myelodysplasia can be explained by a single mis-splicing event. Relative lack of alterations in established driver genes rather supports the previously proposed concept that multiple splicing alterations may cooperatively contribute to the pathogenesis of MDS¹⁶. This is paralleled by the molecular pathogenesis of the myelodysplastic syndrome with deletion 5q, in which haploinsufficiency of a combination of key genes mapping to the commonly deleted region results in a specific myelodysplastic phenotype⁴⁷.

Point 2. The legend for Figure 5a requires some clarification. My interpretation is that the bars represent SF3B1-associated aberrant splicing detected in patient samples and the shading reflects the fraction with support in the engineered cell line. ‘Validated’ is somewhat misleading, since this generally implies confirmation using orthogonal technologies. In panels b and c, the splicing changes are qualitatively recapitulated in the cell lines, but it would be useful to know how consistent the results were at the individual transcript level. Perhaps a supplemental figure could show delta PSI values for aberrant transcripts in patient samples vs. cell lines.

As the reviewer mentioned, the red shading in Figure 5a indicates alternative splicing events that showed a consistent change with P value <0.05 in the CRISPR cell lines (*SF3B1* wild-type lines [N=3] and *SF3B1* mutant lines [N=3]). Due to the small number of independent cell lines, not all the events reached statistical significance. We agree with the reviewer that the word ‘validated’ is misleading. We have modified

the name for Figure 5, Figure 5a, and its legend as follows. We also avoided use of the word ‘validated’ in the main text.

Page 43, in Line 1

Figure 5. Recapitulation of mutant *SF3B1*-associated abnormal splicing *in vitro*

(a) The number of mutant *SF3B1*-associated events that showed a consistently significant difference in CRISPR cell lines with the *SF3B1*^{K700E} mutation. Bars on the right indicate alternative splicing events that were more frequently found in *SF3B1*-mutated cell lines. Bars on the left represent those that were more frequently found in the controls. Red bars indicate events with consistent changes with a *P* value <0.05 in CRISPR cell lines.

We also showed delta PSI values for aberrant transcripts in primary samples vs. CRISPR cell lines in Supplementary Figure 18.

Point 3. The RT-PCR image quality is not adequate in Figure 6. The results would also be more convincing with quantitative analysis of transcript ratios.

We deeply appreciate the reviewer’s advice. In the previous manuscript, we had performed little modification of the RT-PCR images. We performed a levels adjustment of the RT-PCR images for clarity. Consistent with our RNA-seq analysis, differences in the amount of intron-retaining isoforms were small. We thus showed the relative intensity of RT-PCR bands in the Figures 6a and 6b, as the reviewer suggested.

Minor comments

Point 1. The PSI scale in Supp Figure 3 is not shaded.

The PSI scale in Supplementary Figure 3 is shaded as shown below in our file. If the shading is lost in the file for the reviewer, we are afraid that this might be a problem during file conversion in the submission process. We would like to consult the editorial office.

Point 2. Are the x-axis labels reversed in Figure 7 panels f and g?

As the reviewer pointed out, the x-axis labels in Figs. 7f and 7g were reversed. We have corrected the labels appropriately.

Point 3: Legend for Figure 8 has panel f mislabeled.

We have corrected the mislabelling.

REVIEWERS' COMMENTS:

Reviewer #2 (Remarks to the Author):

The authors have adequately addressed the previous criticisms.

Reviewer #2

- 1. The authors have adequately addressed the previous criticisms.**

We deeply appreciate the reviewer's help to improve our paper.